# Does Chronic Kidney Disease Facilitate Malignant Myocardial Fibrosis in Heart Failure with Preserved Ejection Fraction of Hypertensive Origin?

**DOI:** 10.3390/jcm9020404

**Published:** 2020-02-03

**Authors:** Rocio Eiros, Gregorio Romero-González, Juan Jose Gavira, Oscar Beloqui, Inmaculada Colina, Manuel Fortún Landecho, Begoña López, Arantxa González, Javier Díez, Susana Ravassa

**Affiliations:** 1Department of Cardiology and Cardiac Surgery, Clínica Universidad de Navarra, 31008 Pamplona, Spain; jjgavira@unav.es (J.J.G.); jadimar@unav.es (J.D.); 2Department of Nephrology, Clínica Universidad de Navarra, 31008 Pamplona, Spain; gromero@unav.es; 3Program of Cardiovascular and Renal Diseases, Instituto de Investigación Sanitaria de Navarra (IdiSNA), 31008 Pamplona, Spain; obeloqui@unav.es (O.B.); icolina@unav.es (I.C.); mflandecho@unav.es (M.F.L.); blopez@unav.es (B.L.); amiqueo@unav.es (A.G.); sravassa@unav.es (S.R.); 4Department of Internal Medicine, Clínica Universidad de Navarra, 31008 Pamplona, Spain; 5Program of Cardiovascular Diseases, CIMA Universidad de Navarra, 31008 Pamplona, Spain; 6Centro de Investigación Biomédica en Red de Enfermedades Cardiovasculares (CIBERCV), 28029 Madrid, Spain

**Keywords:** arterial hypertension, heart failure, chronic kidney disease, myocardial fibrosis, biomarkers

## Abstract

In hypertensive patients with heart failure (HF) a serum biomarker combination of high carboxy-terminal propeptide of procollagen type-I (PICP) and low carboxy-terminal telopeptide of collagen type-I to matrix metalloproteinase-1 (CITP:MMP-1) ratio identifies a histomolecular phenotype of malignant myocardial fibrosis (mMF) associated with severe diastolic dysfunction (DD) and poor outcomes. As chronic kidney disease (CKD) facilitates MF and DD, we investigated the influence of CKD on the mMF biomarker combination in HF patients with preserved ejection fraction (HFpEF). Hypertensives (*n* = 365), 232 non-HF and 133 HFpEF, were studied, and 35% non-HF and 46% HFpEF patients had CKD (estimated glomerular filtration rate < 60 mL/min/1.73 m^2^ or urine albumin-to-creatinine ratio ≥ 30 mg/g). Specific immunoassays were performed to determine biomarkers. Medians were used to establish the high PICP and low CITP:MMP-1 combination. A comparison with non-HF showed that the biomarker combination presence was increased in HFpEF patients, being associated with CKD in all patients. CKD influenced the association of the biomarker combination and HFpEF (*p* for interaction ≤ 0.019). The E:e’ ratio was associated with the biomarker combination in CKD patients. Among CKD patients with HFpEF, those with the biomarker combination exhibited higher (*p* = 0.016) E:e’ ratio than those without the pattern. These findings suggest that CKD facilitates the development of biomarker-assessed mMF and DD in hypertensive HFpEF patients.

## 1. Introduction

Diffuse myocardial fibrosis (MF) is practically a constant finding in hypertensive patients with heart failure (HF) and contributes to left ventricular (LV) stiffness and dysfunction [1]. In hypertensive patients with HF, MF is the result of increased deposition of collagen type I fibers with increased cross-linking [1]. In HF patients the amount of collagen deposition is associated with all-cause death [2], whereas in HF hypertensive patients the degree of collagen cross-linking is associated with the risk of hospitalization for HF [3]. In fact, the combination of extensive collagen deposition and excessive collagen cross-linking identifies a histomolecular phenotype of ”malignant” MF (mMF) that is independently associated with severe LV dysfunction, and both hospitalization for HF and cardiovascular mortality in hypertensive patients with HF [4].

Some circulating biomarkers of myocardial collagen type I deposition and cross-linking have been characterized in HF patients. On the one hand, serum levels of the carboxy-terminal propeptide of procollagen type I (PICP) directly correlate with the amount of collagen type I deposition in the myocardium of hypertensive patients with HF [5]. On the other hand, the serum carboxy-terminal telopeptide of collagen type I to matrix metalloproteinase-1 ratio (CITP:MMP-1) inversely correlates with myocardial collagen type I cross-linking in hypertensive patients with HF [3]. Recently, it has been reported that in hypertensive patients with HF a combination of high PICP and low CITP:MMP-1 ratio identifies a subgroup of patients carrying the histomolecular phenotype of mMF and presenting with high risk of HF hospitalization or death from cardiovascular causes [4], and atrial fibrillation [6].

The presence of chronic kidney disease (CKD) is associated with a higher risk of mortality in HF patients, showing a greater prognostic significance among those with HF with preserved ejection fraction (HFpEF) [7]. In fact, a recent study has shown that CKD is related to cardiac remodeling, significantly impaired cardiac function and, worse outcomes in HFpEF patients, of whom > 80% were hypertensive [8]. Several post-mortem studies have shown MF in patients with CKD [9,10,11], and it has been proposed that the development of LV diastolic dysfunction (LVDD) and subsequent HFpEF in CKD patients is related to MF [12]. Although some studies have explored MF in patients with end-stage CKD using circulating collagen-related biomarkers [13,14], no information is available on serum PICP and CITP/MMP-1 ratio in CKD patients with HF. Therefore, this study was designed to assess in hypertensive patients with HFpEF the potential influence of CKD on the biomarker combination corresponding to the histomolecular phenotype of mMF and the association of this combination with LVDD in these patients.

## 2. Material and Methods

### 2.1. Study Subjects

Samples and data from patients included in the study were provided by the Biobank of the University of Navarra and were processed following standard operating procedures approved by the Clinical Investigation Ethics Committee of the University of Navarra. All subjects gave written informed consent to participate in the study. The study conformed to the principles of the Helsinki Declaration of 1975, as revised in 2013.

In total, 411 patients, diagnosed with chronic arterial hypertension, were enrolled between October 2014 and November 2017, at the University of Navarra Clinic. Blood pressure measurement was obtained by using an appropriate cuff size for the left arm circumference with the patient in a sitting position; after a 5 min rest, measurements were taken twice with a 1 to 2 min interval between the measurements. Hypertension was defined as systolic blood pressure >139 mm Hg and/or diastolic blood pressure > 89 mm Hg and/or antihypertensive treatment. All patients underwent appropriate clinical and laboratory evaluation to exclude secondary hypertension. Of the 411 patients diagnosed as hypertensives, 20 did not have available blood samples and 26 had undetectable or above the range (determined by the respective standard curves) PICP, CITP, or MMP-1 values. Therefore, valid determinations of the biomarkers of myocardial fibrosis were obtained in 365 patients. These were divided in 2 groups, according to the absence of HF (*n* = 232) or presence of HFpEF (*n* = 133). HFpEF was defined by past or current symptoms of HF associated with EF values ≥50%, relevant structural heart disease and/or LVDD, and plasma values of amino-terminal propeptide of brain natriuretic peptide (NT-proBNP) > 125 pg/Ml [15]. CKD was defined on the basis of reduced estimated glomerular filtration rate (eGFR <60 mL/min/1.73 m^2^) or a urine albumin-to-creatinine ratio ≥30 mg/g for ≥3 months [16].

Comorbidities were defined as follows: Obesity was defined as body mass index > 30 kg/m^2^. Dyslipemia was diagnosed if the fasting serum total cholesterol was ≥ 200 mg/dL or serum triglyceride levels were ≥ 150 mg/dL. Diabetes mellitus was defined by physician-documented history, use of oral hypoglycemic or insulin for the treatment of hyperglycemia. Obstructive sleep apnea hypopnea syndrome was diagnosed if the apnea hypopnea index was ≥ 4.6. Chronic obstructive pulmonary disease was diagnosed if the ratio of post-bronchodilator forced expiratory volume in 1 second and forced vital capacity was < 0.70. Anemia was diagnosed if the value of hemoglobin was < 13 g/dL for men and < 12 g/dL for women.

Patients with LVEF values < 50%, severe valvular heart disease, ischemic heart disease, and stage 5 CKD (eGFR < 15 ml/min per 1.73 m^2^) were excluded after examination. None of the patients presented extracardiac conditions associated with alterations in serum or plasma levels of any of the studied collagen biomarkers (i.e., chronic liver disease or metabolic bone disease).

### 2.2. Echocardiographic Study

Two dimensional echocardiographic, pulsed Doppler, and tissue Doppler imaging studies were performed in all patients. LVH was diagnosed when the LV mass index (LVMI) was > 125 g/m^2^ for men and > 95 g/m^2^ for women) [17]. Values of left atrial volume index (LAVI) > 34 mL/m^2^ were considered indicative of LA enlargement [17]. Values of the peak early diastolic velocity (E) to the early mitral annulus velocity in diastole (septal and lateral average) (e’) (E:e’) ratio > 15 were considered indicative of LV diastolic dysfunction [17].

### 2.3. Biochemical Determinations

Venous blood samples were obtained in each patient included in the study from the left antecubital vein and stored at −20 °C for further simultaneous processing. Determination of biomarkers was performed as previously described [4]. Plasma amino-terminal propeptide of brain natriuretic peptide (NT-proBNP) was measured using an ELISA (Roche Diagnostics, Indianapolis, IN, USA). The inter-assay and intra-assay coefficients of variation were less than 10%. The lower limit of detection was 5 pg of NT-proBNP per mL. Serum carboxy-terminal telopeptide of collagen type I (CITP) was measured by an ELISA (Orion Diagnostica, Espoo, Finland). The inter-assay and intra-assay coefficients of variation were 10.3% and 8.8%, respectively. The lower limit of detection was 0.3 µg of CITP per liter. Total serum matrix metalloproteinase-1 (MMP-1) was measured by an alphaLISA (PerkinElmer, Waltham, MA, USA). The inter-assay and intra-assay coefficients of variation were 12.5% and 4%, respectively. The lower limit of detection was 82.6 pg/mL. CITP and MMP-1 values were expressed in molarity and their ratio was calculated in each patient as previously reported [3]. Serum PICP was measured using the EIA MicroVue CICP (Quidel Corporation, San Diego, CA, USA). The inter-assay and intra-assay coefficients of variation were 12.0% and 8.1%, respectively. The lower limit of detection was 0.2 ng/mL.

### 2.4. Statistical Analysis

The presence of the biomarker combination of high PICP and low CITP:MMP-1 ratio was determined by using the previously defined cut-off points [4] and the respective medians in the entire group. Normality was demonstrated by the Shapiro–Wilks or Kolmogorov–Smirnov tests. Non-normally distributed variables were examined after logarithmic transformation. Differences between two groups of subjects were tested by Student’s t-test for unpaired data once normality was demonstrated; otherwise, the nonparametric test (Mann–Whitney U test) was used. Differences between more than two groups were tested by using on-way analysis of variance (ANOVA) followed by the Fisher test once normality was demonstrated; otherwise the nonparametric test of Kruskal–Wallis was applied followed by the Mann–Whitney U test. Categorical variables were examined by using χ^2^ test or Fisher’s exact test, when necessary. Multivariable logistic regression models were used to assess the independent relationships of comorbidities with the presence or absence of the biomarker combination after adjustment for relevant covariables identified by a backward stepwise selection with minimization of the Akaike information criterion (AIC). Calibration of the logistic models was assessed using the Homer–Lemeshow goodness-of-fit test.

To determine whether the association of the biomarker combination with HFpEF differed by the presence or absence of CKD, qualitative interaction analyses were performed by logistic regression analyses in a model including the combination of biomarkers (yes/no) as the dependent variable, and CKD (yes/no), HFpEF (yes/no), and an interaction term derived from the product of these variables as independent categorical factors.

Multivariable linear regression models were performed to assess the independent relationship of the biomarker combination with the E:e’ ratio, adjusting for covariables significant in univariable analyses (*p* < 0.05 in all CKD patients, *p* < 0.1 in CKD patients with HFpEF). The assumption of normality of residuals was checked by the Kolmogorov–Smirnov test and graphic analysis of the P-P plots. Multicollinearity was defined as variance inflation factor (VIF) > 2 or tolerance < 0.50 with model reduction, in case any variable showed evidence of multicollinearity. Age- and sex-adjusted analyses of variance were used to compare the E:e’ ratio among patients classified in four subgroups according to the presence or absence of HFpEF and the biomarker combination.

Values are expressed as mean ± SD or median (interquartile range), and categorical variables as numbers and percentages. Statistical significance was set as a 2-sided *p* of 0.05. The statistical analyses were performed by using SPSS (15.0 version, Chicago, IL, USA) and STATA (12.1 version, Stata Corp, College station, TX, USA) software.

### 2.5. Data Availability

The used study data are unsuitable for public deposition due to ethical restrictions and privacy of participant data. Data are available from these studies for any interested researcher who meets the criteria for access to confidential data. Rocio Eiros can be contacted to request study data.

## 3. Results

### 3.1. Findings in Patients Classified According to the Absence of HF or Presence of HFpEF

Clinical, biochemical, and echocardiographic characteristics of the two groups, non-HF and HFpEF patients, are presented in Table 1. Among other significant clinical differences, CKD was more prevalent in HFpEF patients than in non-HF patients. As expected, more HFpEF patients exhibited abnormal values of the E:e’ ratio and left atrial volume index (LAVI) as compared with non-HF patients. Serum PICP levels were higher and CITP:MMP-1 ratio values lower in HFpEF patients compared with non-HF patients. When the biomarker combination was defined according to the previously defined cut-off points [4], the frequency of patients exhibiting the combination of high PICP (≥111 ng/mL) and low CITP:MMP-1 ratio (≤1.97) was higher (*p* < 0.001) in the HFpEF group (10.5%) as compared with the non-HF group (0.4%). However, since the previously defined biomarker combination was only present in one non-HF patient, subsequent analyses in all patients were performed with the median-defined biomarker combination. As shown in Figure 1A, the frequency of patients exhibiting the combination of high PICP (≥ median value = 70.4 ng/mL) and low CITP:MMP-1 ratio (≤ median value = 3.58) was higher (*p* < 0.001) in the HFpEF group (43.6%) than in the non-HF group (16.4%).

Unadjusted association analyses showed that the median-based biomarker combination of high PICP and low CITP:MMP-1 ratio was associated with several clinical variables in all patients (Table 2). Among the comorbidities, the biomarker panel was associated with dyslipidemia, anemia, and CKD (Table 2). Taking into account all significant univariate associations, a basal model including the variables age, sex, cerebrovascular disease, atrial fibrillation, NYHA class, dyslipidemia, CKD, anemia, and treatment with MR blockers was selected following a backward stepwise selection procedure. Multiple logistic regression analyses showed that the association of the biomarker combination with anemia and CKD were independent of the other previously mentioned covariables (Table 2).

Since CKD and anemia were independently associated with the combination of high PICP and low CITP:MMP-1 ratio and showed a higher prevalence in HFpEF patients, we investigated whether the association of the biomarker combination with HFpEF could be influenced by these factors. Qualitative interaction analyses showed that the prevalence of the combination of high PICP and low CITP:MMP-1 ratio in HFpEF patients was increased by CKD (P for interaction = 0.019). In fact, whereas the biomarker combination was present in 30.6% non-CKD patients with HFpEF (as compared with 16.6% non-CKD patients without HF), its prevalence increased up to 59.0% in CKD patients with HFpEF (as compared with 16.0% in CKD patients without HF) (Figure 1B). The interaction analysis with anemia was nonsignificant (P for interaction = 0.18). In addition, interaction analyses with other demographic factors and comorbidities showed non-significant *p* interaction values. 

### 3.2. Findings in Patients Classified According to the Presence or Absence of CKD and Subcategorized According to the Absence of HF or the Presence of HFpEF

Clinical, biochemical, and echocardiographic characteristics of the four subgroups of patients are presented in Table 3. Of note, abnormal values of E:e’ ratio and LAVI were more frequent in HFpEF patients as compared with non-HF patients, irrespectively of the presence or absence of CKD (*p* < 0.001). Interestingly, the increment in both E:e’ ratio and LAVI was higher in CKD patients with HFpEF as compared with non-CKD patients with HFpEF (*p* ≤ 0.050, Table 3).

Serum PICP levels were increased in HFpEF patients compared with non-HF, irrespectively of the presence or absence of CKD. In addition, PICP was higher in CKD patients with HFpEF as compared with non-CKD patients with HFpEF (Table 3). Whereas no differences in the CITP:MMP-1 ratio were observed between non-CKD patients with and without HFpEF, this parameter was decreased in CKD patients with HFpEF as compared with non-HF patients (Table 3). As previously mentioned, the frequency of patients exhibiting the combination of high PICP and low CITP:MMP-1 ratio was higher (*p* < 0.001) in CKD patients with HFpEF than in CKD patients without HF and non-CKD patients with HFpEF (Figure 1B). 

The associations of the biomarker combination with the E:e’ ratio and LAVI were first analyzed in all patients, categorized according to the presence or absence of CKD. Non-CKD patients with the biomarker combination (*n* = 47, 21.0%) exhibited similar E:e’ ratio and LAVI values as compared with non-CKD patients without the combination (E:e’ ratio 10.3 ± 3.5 vs. 9.3 ± 3.4, *p* = 0.09 and LAVI 29.4 ± 12.0 vs. 27.2 ± 10.1 mL/m^2^, *p* = 0.25). CKD patients with the biomarker combination (*n* = 49, 34.5%) exhibited higher E:e’ ratio as compared with CKD patients without the combination (13.8 ± 5.1 vs. 8.9 ± 4.0, *p* < 0.001), whereas LAVI values were similar in the two group of patients (33.8 ± 13.9 vs. 29.1 ± 11.3 mL/m^2^, *p* = 0.08). The association of the biomarker combination with higher values of E:e’ in CKD patients was independent of confounding factors (Table 4). Interestingly, this association was also present in the subgroup of CKD patients with HFpEF, and was independent of a basal model including sex, heart rate, NYHA class, and NT-proBNP (Table 5). In these patients, the proportion of variance in the E:e’ variable explained by the basal model (R2) was 39.8%, which was increased up to 44.9% (*p* = 0.033) by the addition of the biomarker combination.

We, thus, decided to analyze the influence of the biomarker combination on the E:e’ ratio in both non-CKD and CKD patients stratified according to the absence or presence of HFpEF. As shown in Figure 2A, age and sex-adjusted analyses showed that in non-CKD patients, those with HFpEF exhibited higher E:e’ ratio values than non-HF patients, irrespectively of the presence of the biomarker combination. However, in CKD patients the increase in E:e’ ratio values observed in HFpEF patients was significantly higher in those presenting with the biomarker combination compared with patients without the combination (Figure 2B).

## 4. Discussion

The main findings here reported are the following: (i) Hypertensive patients with HFpEF exhibit increased serum PICP levels and decreased serum CITP:MMP-1 ratio values as compared with hypertensive patients without HF; (ii) the prevalence of a combination of high serum PICP and low serum CITP:MMP-1 ratio is higher in hypertensive patients with HFpEF than in hypertensive patients without HF; (iii) the presence of CKD, but not of other comorbidities, enhances the association of the biomarker combination with HFpEF; and (iv) the biomarker combination associates with LVDD in patients with CKD, especially in those with HFpEF. Collectively, these findings suggest that alterations in collagen type I metabolism, as assessed by the biomarkers PICP and CITP:MMP-1 ratio, worsen with the transition to HFpEF in hypertensive patients, and that CKD facilitates the development of biomarker-assessed mMF and LVDD in hypertensive patients with HFpEF.

PICP is formed during the extracellular conversion of procollagen type I into mature fibril-forming collagen type I by the enzyme procollagen carboxy-terminal proteinase or bone morphogenetic protein-1 (PCP/BMP-1) and can reach the blood stream through tissue capillaries [18]. Several clinical studies have provided evidence that serum PICP is directly correlated with the amount of collagen deposition, namely collagen type I, in MF associated with HF attributable to arterial hypertension [5,19,20]. Therefore, it is tempting to speculate that the excess of serum PICP in hypertensive patients with HFpEF, as compared with hypertensive patients without HF here reported, could reflect an increase in myocardial synthesis and deposition of collagen type I fibers in parallel with the progression towards symptomatic HF in these patients. 

The degree of collagen cross-linking is regulated by several enzymes, namely lysyl oxidase (LOX), and determines the resistance of collagen type I fibers to degradation by MMP-1, resulting in diminished cleavage of CITP [21]. As previously mentioned, we have shown that the serum CITP:MMP-1 ratio is inversely correlated with myocardial collagen cross-linking in patients with HF due to hypertension [3]. Therefore, the lower CITP:MMP-1 ratio in patients with HFpEF as compared with patients without HF here observed suggests that the cross-linking among collagen type I fibrils increases with the worsening of cardiac function in hypertensive patients.

Recently, we reported a histomolecular phenotype of MF characterized by the coincidence of extensive collagen type I deposition and excessive cross-linking in endomyocardial biopsies from one third of hypertensive patients with HF [4]. In the same study, we found that the combination of high serum PICP and low serum CITP:MMP-1 ratio was indicative of the concurrence of severe myocardial collagen type I deposition and abnormally high myocardial collagen cross-linking, respectively, in these patients [4]. The concurrence of extensive myocardial collagen type I deposition and excessive cross-linking in hypertensive patients with HF is associated with elevated filling pressures and mechanical stress of the failing left ventricle, as well as with HF hospitalization after enrollment or death from cardiovascular causes [4]. Furthermore, the biomarker combination of high serum PICP and low serum CITP:MMP-1 ratio is associated also with increased risk of HF hospitalization and mortality [4], as well as with atrial fibrillation [6]. Therefore, we proposed that in hypertensive patients with HF the biomarker combination identifies a histomolecular phenotype of MF with severe LV dysfunction and poor prognosis (i.e., mMF). In this study, by performing a median-based categorization we show that the prevalence of the biomarker combination increases two times in hypertensive patients with HFpEF and absence of CKD and up to four times in hypertensive patients with HFpEF and CKD as compared with their respective groups of hypertensive patients without HF. Therefore, CKD emerges as a comorbidity that could facilitate the development of mMF in HFpEF. This possibility could explain why CKD is independently associated with abnormal LV mechanics and adverse outcomes in patients with HFpEF [8].

Because of the high prevalence of MF in patients with CKD and the direct correlation existing between the degree of impairment of kidney function and the severity of MF [9,10,11], it has been proposed that pro-fibrotic factors or pathways linked to CKD could exist [22]. Although we did not investigated mechanisms of mMF in the current study, some experimental and clinical evidence point to the matricellular protein osteopontin as a potential candidate. On the one hand, osteopontin mediates the myocardial fibrotic response in experimental pressure overload [23] and associates with collagen type I deposition and cross-linking in hypertensive patients with HF [24], and patients with mMF exhibit the highest expression of osteopontin in the myocardium [25]. On the other hand, osteopontin predicts HF hospitalization and mortality in patients with HFpEF [22]. Finally, plasma osteopontin progressively increases with the decline in eGFR in patients with CKD [26] and is associated with cardiovascular mortality in these patients [27]. Further clinical studies are required to delineate the role of osteopontin in the development of mMF in CKD patients with HFpEF.

From the point of view of collagen type I metabolism, the inhibition of the PCP/BMP-1-LOX axis has been proposed as a potential tailored antifibrotic therapy in patients with mMF. In this regard, administration of torasemide in addition to standard HF therapy has been associated with reductions in myocardial collagen deposition and cross-linking in conjunction with decreased activation of PCP/BMP-1 and diminished expression of LOX [20,28,29]. Additionally, 80% of the torasemide-treated patients exhibited normalization of LV stiffness and improvement of LV function [29]. Of note, none of these effects were observed in furosemide-treated HF patients [20,28,29]. Whether CKD patients with HFpEF carrying the combination of high serum PICP and low serum CITP:MMP-1 ratio would benefit specifically from the antifibrotic properties of torasemide remains to be tested in an adequately designed trial.

Some limitations need to be acknowledged. First, the present study was a single-center transversal study with a small number of patients, which could have caused selection bias. Second, results here presented cannot be extrapolated to non-hypertensive patients with HF or to hypertensive patients with HF and reduced or mid-range EF, or to patients with HFpEF with stage 5 CKD. Third, in this study, the presence of the biomarker combination has been analyzed by using the PICP and CITP:MMP1 medians rather than the cut-off points previously defined [4], since levels of PICP were lower and CITP:MMP1 values were higher in the patients here analyzed, suggesting the presence of a less severe myocardial fibrosis in terms of the quantity and quality of the collagen fiber as compared with the previously studied HF patients (including HFpEF and HF with reduced EF) [4]. As a consequence, the low frequency of the biomarker combination as defined in previous studies precluded the performance of subgroup analyses in the current study. In this regard, further studies should be performed to examine the correspondence between histological and biochemical aspects of MIF in different HF stages. Fourth, subgroup analyses are commonly considered as exploratory analyses with limited generalizability. Therefore, further analyses in large and independent cohorts of patients are necessary to confirm these findings. Fifth, potential problems related to multiplicity could have influenced the findings obtained. Finally, because they are descriptive in nature, the associations found between renal disease, circulating biomarkers, and LV dysfunction do not establish causality.

## 5. Conclusions

In conclusion, the prevalence of a biomarker combination of high serum PICP and low serum CITP:MMP-1 ratio that identifies a histomolecular phenotype of mMF is associated with CKD in hypertensive patients with HFpEF. In addition, there is an effect modification of CKD on the association of this biomarker combination with LVDD in HFpEF patients. There is a necessity to investigate the pathophysiological mechanisms linking the increased prevalence of the biomarker combination with CKD, as well as its association with poor outcome in larger cohorts of hypertensive patients with HFpEF, with and without CKD. The possibility exists that the biomarker combination could serve to select hypertensive patients with HFpEF and CKD subsidiary of a tailored treatment with torasemide on top of other HF medications.

## Figures and Tables

**Figure 1 jcm-09-00404-f001:**
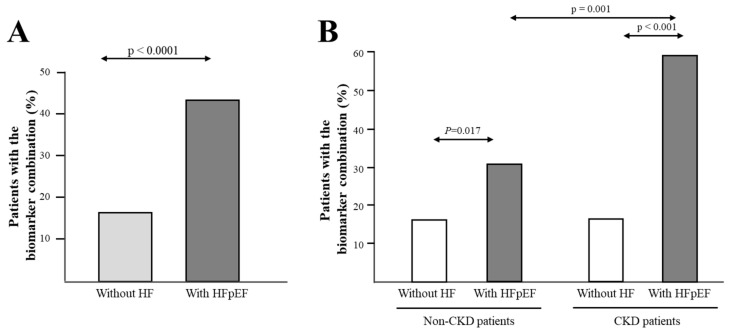
Frequency distribution of the combination of biomarkers of malignant myocardial fibrosis as defined in the text. (**A**) Panel A shows frequency distribution in all hypertensive patients classified according to the absence of heart failure (HF) or the presence of HF with preserved ejection fraction (HFpEF); (**B**) Panel B shows frequency distribution in patients without and with chronic kidney disease (non-CKD and CKD groups, respectively) classified in subgroups according to the absence of HF or the presence of HFpEF.

**Figure 2 jcm-09-00404-f002:**
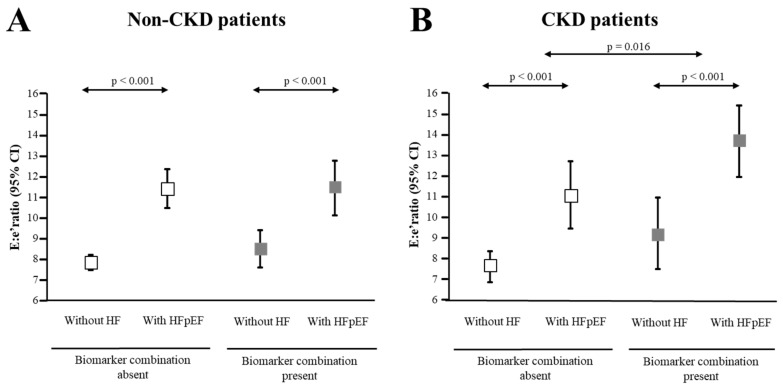
Distribution of the maximal early transmittal flow velocity in diastole (E) to the early mitral annulus velocity in diastole (septal and lateral average) (e’) (E:e’) ratio values. (**A**) Panel A shows values in the group of patients without chronic kidney disease (non-CKD patients); (**B**) Panel B shows values in the group of patients with CKD (CKD patients). Patients were further classified according to the presence or absence of the combination of biomarkers of malignant myocardial fibrosis as defined in the text and then to the absence of heart failure (without HF) or the presence of HF with preserved ejection fraction (with HFpEF). Symbols represent means and 95% confidence intervals adjusted for age and sex.

**Table 1 jcm-09-00404-t001:** Characteristics of patients classified according to the presence or absence of preserved ejection fraction (HFpEF).

	Without HF (*N* = 232)	With HF (*N* = 133)	*p* Value
Age, years	62.3 ± 9.7	74.0 ± 7.7	<0.001
Male, *n* (%)	169 (72.8)	43 (32.3)	<0.001
BMI, kg/m^2^	29.2 ± 4.6	29.1 ± 4.2	0.90
SBP, mmHg	135 ± 18.3	136 ± 20.3	0.58
DBP, mmHg	81.8 ± 10.4	74.9 ± 12.0	<0.001
MAP, mmHg	99.4 ± 11.5	95.2 ± 13.5	0.003
HR, beats/min	66.5 ± 11.3	68.9 ± 14.9	0.12
Previous cardiovascular history, *n* (%)			
Hospitalized within 12 months	0 (0.0)	46 (34.3)	
Peripheral artery disease	2 (0.9)	5 (3.8)	0.10
Cerebrovascular disease	2 (0.9)	9 (6.8)	0.002
Atrial fibrillation	8 (3.4)	48 (36.1)	<0.001
NYHA class			
I		29 (21.8)	
II		70 (52.6)	
III		34 (25.6)	
Comorbidities, *n* (%)			
Obesity	92 (39.7)	51 (38.3)	0.81
Dyslipidemia	110 (47.4)	87 (65.4)	0.001
Diabetes	40 (17.2)	27 (20.3)	0.47
OSAHS	7 (3.0)	11 (8.3)	0.041
COPD	0 (0.0)	8 (6.0)	
Anemia	14 (6.0)	23 (17.3)	0.001
CKD	81 (34.9)	61 (45.9)	0.039
Treatments, *n* (%)			
Beta-blockers	40 (17.2)	101 (75.9)	<0.001
ACEI/ARB	164 (70.7)	106 (79.7)	0.06
Diuretics	87 (37.5)	100 (75.2)	<0.001
MR blockers	9 (3.9)	50 (37.6)	<0.001
Anti-diabetic drugs	35 (15.1)	25 (18.8)	0.36
Biochemical parameters			
ACR, mg/g	8.1 (4.7–15.8)	18.4 (8.4–34.8)	<0.001
eGFR, mL/min/1.73 m^2^	76.9 ± 21.5	62.7 ± 18.9	<0.001
Hemoglobin, g/dL	14.8 ± 1.4	13.2 ± 1.6	<0.001
NT-proBNP, pg/mL		332 (191–782)	
PICP, ng/mL	61.1 (50.2–79.5)	91.0 (70.6–108)	<0.001
CITP:MMP-1 ratio	4.0 (2.3-6.6)	3.0 (1.5–5.0)	<0.001
Echocardiographic parameters			
LV morphology			
LVMI, g/m^2^	112 ± 30.7	119 ± 30.1	0.056
LVH, *n* (%)	116 (50.0)	94 (70.7)	<0.001
RWT	0.41 ± 0.08	0.47 ± 0.10	<0.001
RWT > 0.45, *n* (%)	60 (25.9)	73 (54.9)	<0.001
LVEDVi, mL/m^2^	58.3 ± 19.9	40.1 ± 13.4	<0.001
LVESVi, mL/m^2^	21.1 ± 8.6	14.2 ± 7.2	<0.001
LV function			
E wave, cm/s	69.5 ± 15.9	82.8 ± 25.4	<0.001
E:A ratio	0.89 ± 0.23	0.95 ± 0.44	0.51
DT, ms	219 ± 67.4	228 ± 65.3	0.22
Mean e ‘, cm/s	9.2 ± 2.6	7.1 ± 2.1	<0.001
E:e’ ratio	8.0 ± 2.7	13.3 ± 4.3	<0.001
E:e’ ratio > 15, *n* (%)	3 (1.3)	37 (27.8)	<0.001
LVEF, %	63.6 ± 6.3	67.4 ± 8.2	<0.001
LA morphology			
LAVI, mL/m^2^	25.4 ± 7.0	33.5 ± 12.5	<0.001
LAVI > 34 mL/m^2^, *n* (%)	24(10.3)	56(42.1)	<0.001

HF means heart failure; BMI, body mass index; SBP, systolic blood pressure; DBP, diastolic blood pressure; MAP, mean arterial pressure; HR, heart rate; NYHA, New York Heart Association; OSAHS, obstructive sleep apnea hypopnea syndrome; COPD; chronic obstructive pulmonary disease; CKD, chronic kidney disease; ACEI, angiotensin converting enzyme inhibitor; ARB, angiotensin II type 1 receptor blockers; MR, mineralocorticoid receptor; ACR, albumin to creatinine ratio; eGFR, estimated glomerular filtration rate; NT-proBNP, N-terminal pro-B-type natriuretic peptide; PICP, carboxy-terminal propeptide of procollagen type I; CITP:MMP-1, carboxy-terminal telopeptide of collagen type I to serum matrix metalloproteinase-1 ratio; LV, left ventricular; LVMI, LV mass index; LVH, LV hypertrophy; RWT, relative wall thickness; LVEDVi, LV end-diastolic volume index; LVESVi, LV end-systolic volume index; E, peak early diastolic velocity; A, peak late diastolic velocity; DT, deceleration time; e’, mean peak early diastolic mitral annular velocity at the septal and lateral acquisition sites; LVEF, LV ejection fraction; LA, left atrial; LAVI, left atrial volume index. Quantitative variables are expressed as mean ± SD or as median (interquartile range). Categorical variables are expressed as numbers (percentages).

**Table 2 jcm-09-00404-t002:** Logistic regression analyses for the biomarker combination.

	Univariable Analyses		Multivariable Analysis	
	OR (95% CI)	*p* Value *	OR (95% CI)	*p* Value
Age, years	1.04 (1.02 to 1.07)	0.001	1.01 (0.98 to 1.04)	0.48
Male, *n* (%)	0.48 (0.30 to 0.77)	0.002	0.64 (0.37 to 1.11)	0.11
BMI, kg/m^2^	0.95 (0.90 to 1.00)	0.07		
SBP, mmHg	1.00 (0.99 to 1.01)	0.96		
DBP, mmHg	0.98 (0.96 to 1.00)	0.06		
MAP, mmHg	0.99 (0.97 to 1.01)	0.25		
HR, beats/min	0.99 (0.97 to 1.01)	0.24		
Previous cardiovascular history, *n* (%)				
Hospitalized within 12 months	2.46 (1.30 to 4.65)	0.006		
Cerebrovascular disease	3.52 (1.05 to 11.8)	0.042	1.97 (0.51 to 7.54)	0.32
Atrial Fibrillation	2.45 (1.36 to 4.44)	0.003	1.46 (0.75 to 2.86)	0.27
NYHA class (II–III)	2.86 (1.74 to 4.69)	<0.0001	1.43 (0.74 to 2.78)	0.29
Comorbidities, *n* (%)				
Obesity	0.71 (0.44 to 1.16)	0.17		
Dyslipidemia	1.81 (1.12 to 2.93)	0.016	1.45 (0.85 to 2.48)	0.17
Diabetes	0.70 (0.37 to 1.32)	0.27		
OSAHS	2.35 (0.90 to 6.15)	0.08		
COPD	2.88 (0.71 to 11.8)	0.14		
Anemia	2.68 (1.34 to 5.36)	0.005	2.24 (1.04 to 4.81)	0.039
CKD	1.97 (1.23 to 3.17)	0.005	1.87 (1.11 to 3.15)	0.018
Treatments, *n* (%)				
Beta blockers	2.25 (1.40 to 3.62)	0.001		
ACEI/ARB	1.25 (0.73 to 2.16)	0.42		
Diuretics	1.86 (1.16 to 3.00)	0.011		
MR blockers	2.43 (1.36 to 4.35)	0.003	1.52 (0.76 to 3.02)	0.24
Antidiabetic drugs	0.74 (0.38 to 1.44)	0.37		

HF means heart failure; BMI, body mass index; SBP, systolic blood pressure; DBP, diastolic blood pressure; MAP, mean arterial pressure; HR, heart rate; eGFR, estimated glomerular filtration rate; NYHA, New York Heart Association; OSAHS, obstructive sleep apnea hypopnea syndrome; COPD; chronic obstructive pulmonary disease; CKD, chronic kidney disease; ACEI, angiotensin converting enzyme inhibitor; ARB, angiotensin II type 1 receptor blockers; MR, mineralocorticoid receptor. * *p* values <0.05 were selected followed by a backward stepwise selection with minimization of the Akaike information criterion (AIC) for the multivariable analysis.

**Table 3 jcm-09-00404-t003:** Characteristics of patients classified according to presence or absence of CKD and with or without HFpEF.

	Non-CKD (*N* = 223)			CKD (*N* = 142)			
	Without HF (*N* = 151)	With HFpEF (*N* = 72)	*p*	Without HF (N = 81)	With HFpEF (*N* = 61)	*p*	*p**
Age, years	62.2 ± 9.7	73.3 ± 7.6	<0.001	62.5 ± 9.5	74.7 ± 7.8	<0.001	0.99
Male, *n* (%)	95 (62.9)	23 (31.9)	<0.001	74 (91.4)	20 (32.8)	<0.001	0.92
BMI, kg/m^2^	28.5 ± 4.1	29.0 ± 4.1	0.99	30.4 ± 5.2	29.3 ± 4.4	0.81	0.99
SBP, mmHg	134 ± 17.6	138 ± 19.4	0.99	135 ± 19.8	133 ± 21.2	0.99	0.90
DBP, mmHg	80.6 ± 9.9	75.9 ± 11.6	0.020	83.9 ± 10.9	73.7 ± 12.6	<0.001	0.71
MAP, mmHg	98.5 ± 10.9	96.6 ± 12.6	0.99	101 ± 12.5	93.5 ± 14.4	0.002	0.87
HR, beats/min	66.1 ± 11.3	69.8 ± 16.3	0.32	67.2 ± 11.3	67.8 ± 13.2	0.99	0.99
Previous cardiovascular history, *n* (%)							
Hospitalized within 12 months	0 (0.0)	25 (34.7)		0 (0.0)	21 (34.4)		0.97
Peripheral artery disease	2 (1.3)	4 (5.6)	0.09	0 (0.0)	1 (1.6)		0.37
Cerebrovascular disease	2 (1.3)	3 (4.2)	0.33	0 (0.0)	6 (9.8)		0.30
Atrial Fibrillation	4 (2.6)	23 (31.9)	<0.001	4 (4.9)	25 (41.0)	<0.001	0.28
NYHA class							
I		16 (22.2)			13 (21.3)		
II		40 (55.6)			30 (49.2)		0.62
III		16 (22.2)			18 (29.5)		
Comorbidities, *n* (%)							
Obesity	56 (37.1)	25 (34.7)	0.73	36 (44.4)	26 (42.6)	0.83	0.35
Dyslipidemia	72 (47.7)	41 (56.9)	0.20	38 (46.9)	46 (75.4)	0.001	0.026
Diabetes	32 (21.2)	13 (18.1)	0.59	8 (9.9)	14 (23.0)	0.033	0.48
OSAHS	6 (4.0)	6 (8.3)	0.21	1 (1.2)	5 (8.2)	0.08	0.98
COPD	0 (0.0)	2 (2.8)		0 (0.0)	6 (9.8)		0.09
Anemia	12 (7.9)	11 (15.3)	0.09	2 (2.5)	12 (19.7)	0.001	0.50
Treatments, *n* (%)							
Beta-blockers	26 (17.2)	55 (76.4)	<0.001	14 (17.3)	46 (75.4)	<0.001	0.90
ACEI/ARB	105 (69.5)	56 (77.8)	0.20	59 (72.8)	50 (82.0)	0.20	0.55
Diuretics	59 (39.1)	49 (68.1)	<0.001	28 (34.6)	51 (83.6)	<0.001	0.039
MR blockers	7 (4.6)	25 (34.7)	<0.001	2 (2.5)	25 (41.0)	<0.001	0.46
Antidiabetic drugs	29 (19.2)	11 (15.3)	0.48	6 (7.4)	14 (23.0)	0.008	0.26
Biochemical parameters							
ACR, mg/g	8.0 (4.9–12.4)	12.7 (5.8–19.3)	0.040	8.5 (4.2–37.8)	37.5 (19.5–47.6)	0.003	<0.001
eGFR, mL/min/1.73m^2^	88.0 ± 15.4	74.2 ± 15.3	<0.001	56.3 ± 15.1	49.3 ± 13.0	0.034	<0.001
Hemoglobin, g/dL	14.6 ± 1.4	13.2 ± 1.4	<0.001	15.1 ± 1.2	13.2 ± 1.8	<0.001	0.99
NT-proBNP, pg/mL		256 (179–502)			464 (199–928)		0.029
PICP, ng/mL	60.4 (50.2–81.7)	81.6 (64.6–96.6)	<0.001	63.9 (50.1–78.1)	106 (88.1–136)	<0.001	<0.001
CITP:MMP-1 ratio	4.1 (2.3–7.3)	3.3 (1.8–5.7)	0.06	4.0 (2.3–6.3)	2.4 (1.2–4.4)	0.004	0.40
Echocardiographic parameters							
LV morphology							
LVMI, g/m^2^	111 ± 32.7	121 ± 26.9	0.08	114 ± 26.6	114 ± 28.0	0.98	0.84
LVH, *n* (%)	75 (49.7)	54 (75.0)	<0.001	41 (50.6)	40 (65.6)	0.08	0.26
RWT	0.40 ± 0.08	0.46 ± 0.09	<0.001	0.41 ± 0.07	0.49 ± 0.11	<0.001	0.035
RWT > 0.45, *n* (%)	40 (26.5)	37 (51.4)	<0.001	20 (24.7)	36 (59.0)	<0.001	0.38
LVEDVi, mL/m^2^	55.8 ± 17.8	40.0 ± 14.8	<0.001	62.9 ± 22.8	40.2 ± 11.7	<0.001	0.96
LVESVi, mL/m^2^	20.2 ± 8.4	14.2 ± 7.5	<0.001	22.7 ± 8.7	14.2 ± 6.9	<0.001	0.99
LV function							
E wave, cm/s	71.1 ± 15.1	79.3 ± 23.1	0.040	66.5 ± 17.0	86.9 ± 27.4	<0.001	0.028
E:A ratio	0.9 ± 0.2	0.9 ± 0.3	0.93	0.9 ± 0.2	1.0 ± 0.6	0.32	0.75
DT, ms	221 ± 62.6	230 ± 64.0	0.36	214 ± 76.3	225 ± 67.1	0.33	0.69
Mean e ’, cm/s	9.3 ± 2.5	7.0 ± 2.0	<0.001	9.1 ± 2.7	6.9 ± 1.7	<0.001	0.80
E:e’ ratio	8.0 ± 2.2	12.5 ± 3.5	<0.001	7.8 ± 2.6	14.2 ± 4.9	<0.001	0.010
E:e’ ratio > 15, *n* (%)	2 (1.3)	15 (20.8)	<0.001	1 (1.2)	22 (36.1)	<0.001	0.050
LVEF, %	64.0 ± 6.3	66.7 ± 7.9	0.050	62.9 ± 6.2	68.2 ± 8.4	<0.001	0.99
LA morphology							
LAVI, mL/m^2^	25.6 ± 7.5	29.7 ± 9.6	0.013	25.0 ± 6.1	38.1 ± 14.1	<0.001	0.001
LAVI > 34 mL/m^2^, n (%)	17 (11.3)	23 (31.9)	<0.001	7 (8.6)	33 (54.1)	<0.001	0.012

HF means heart failure; BMI, body mass index; SBP, systolic blood pressure; DBP, diastolic blood pressure; MAP, mean arterial pressure; HR, heart rate; NYHA, New York Heart Association; OSAHS, obstructive sleep apnea hypopnea syndrome; COPD; chronic obstructive pulmonary disease; CKD, chronic kidney disease; ACEI, angiotensin converting enzyme inhibitor; ARB, angiotensin II type 1 receptor blockers; MR, mineralocorticoid receptor; ACR, albumin to creatinine ratio; eGFR, estimated glomerular filtration rate; NT-proBNP, N-terminal pro-B-type natriuretic peptide; PICP, carboxy-terminal propeptide of procollagen type I; CITP:MMP-1, carboxy-terminal telopeptide of collagen type I to serum matrix metalloproteinase-1 ratio; LV, left ventricular; LVMI, LV mass index; LVH, LV hypertrophy; RWT, relative wall thickness; LVEDVi, LV end-diastolic volume index; LVESVi, LV end-systolic volume index; E, peak-early diastolic velocity; A, peak-late diastolic velocity; DT, deceleration time; e’, mean peak-early diastolic mitral annular velocity at the septal and lateral acquisition sites; LVEF, LV ejection fraction; LA, left atrial; LAVI, left atrial volume index. Quantitative variables are expressed as mean ± SD or as median (interquartile range). Categorical variables are expressed as numbers (percentages). * For comparisons between patients with HFpEF.

**Table 4 jcm-09-00404-t004:** Linear regression analyses for the E:e’ ratio in CKD patients.

	Univariable Analyses		Multivariable Analysis		
	Estimate (95%CI)	*p**	Estimate (95%CI)	Partial R^2^ (%)	*p*
Age, years	0.21 (0.14 to 0.28)	<0.001	0.06 (−0.02 to 0.14)	0.98	0.13
Male, (no = 0, yes = 1)	−5.58 (−7.07 to −4.10)	<0.001	−2.27 (−4.01 to −0.52)	2.76	0.011
BMI, kg/m^2^	0.04 (−0.13 to 0.21)	0.65			
SBP, mmHg	−0.003 (−0.04 to 0.04)	0.88			
DBP, mmHg	−0.10 (−0.16 to −0.03)	0.003	0.02 (−0.05 to 0.08)	0.10	0.64
HR, beats/min	−0.08 (−0.16 to −0.01)	0.026	−0.07 (−0.13 to −0.01)	2.50	0.017
ACR (log_2_), mg/g	0.32 (−0.05 to 0.70)	0.09			
eGFR, mL/min/1.73m^2^	−0.05 (−0.10 to 0.009)	0.10			
Hospitalized within 12 months, (no = 0, yes = 1)	4.30 (2.08 to 6.53)	<0.001	−0.48 (−2.62 to 1.65)	0.08	0.65
Cerebrovascular disease, (no = 0, yes = 1)	4.23 (−0.11 to 8.34)	0.054			
Peripheral artery disease, (no = 0, yes = 1)	−1.80 (−6.76 to 3.16)	0.47			
Atrial Fibrillation, (no = 0, yes = 1)	3.25 (1.26 to 5.24)	0.002	0.80 (−0.95 to 2.56)	0.35	0.37
NYHA class (II-III), (no = 0, yes = 1)	6.58 (5.20 to 7.97)	<0.001	3.01 (0.94 to 5.07)	3.50	0.005
Obesity, (no = 0, yes = 1)	0.83 (−0.85 to 2.50)	0.33			
Dyslipidemia, (no = 0, yes = 1)	2.13 (0.47 to 3.79)	0.012	−0.01 (−1.46 to 1.44)	<0.01	0.99
Diabetes, (no = 0, yes = 1)	1.40 (−0.89 to 3.68)	0.23			
OSAHS, (no = 0, yes = 1)	2.12 (−2.00 to 6.23)	0.31			
COPD, (no = 0, yes = 1)	2.40 (−1.72 to 6.51)	0.25			
Anemia, (no = 0, yes = 1)	3.75 (1.04 to 6.47)	0.007	−0.14 (−2.47 to 2.19)	0.01	0.90
Beta-blockers, (no = 0, yes = 1)	4.97 (3.51 to 6.44)	<0.001	1.12 (- 0.55 to 2.80)	0.74	0.19
ACEI/ARB, (no = 0, yes = 1)	0.27 (−1.70 to 2.24)	0.79			
Diuretics, (no = 0, yes = 1)	4.00 (2.45 to 5.53)	<0.001	0.17 (−1.47 to 1.81)	0.02	0.84
MR blockers, (no = 0, yes = 1)	5.25 (3.32 to 7.18)	<0.001	1.33 (−0.53 to 3.19)	0.83	0.16
Biomarker combination, (no = 0, yes = 1)	4.98 (3.43 to 6.53)	<0.001	2.04 (0.51 to 3.56)	2.92	0.009

Abbreviations as in Table 1. * Variables with *p* < 0.05 were selected for the multivariable analysis.

**Table 5 jcm-09-00404-t005:** Linear regression analyses for the E:e’ ratio in CKD patients with HFpEF.

	Univariable Analyses		Multivariable Analysis		
	Estimate (95%CI)	*p**	Estimate (95%CI)	Partial R^2^ (%)	*p*
Age, years	0.01 (- 0.15 to 0.18)	0.87			
Male, (no = 0, yes = 1)	−2.60 (−5.26 to 0.05)	0.055	−2.23 (−4.46 to 0.001)	4.28	0.050
BMI, kg/m^2^	0.09 (−0.21 to 0.38)	0.55			
SBP, mmHg	−0.01 (−0.07 to 0.05)	0.67			
DBP, mmHg	0.003 (−0.10 to 0.11)	0.96			
HR, beats/min	−0.14 (−0.23 to −0.05)	0.003	−0.15 (−0.23 to −0.07)	13.8	0.001
ACR (log_2_), mg/g	−0.48 (−1.76 to 0.80)	0.45			
eGFR, mL/min/1.73m^2^	−0.03 (−0.12 to 0.07)	0.62			
NT-proBNP (log_2_), pg/mL	4.15 (1.60 to 6.70)	0.002	3.95 (1.74 to 6.16)	13.6	0.001
Hospitalized within 12 months, (no = 0, yes = 1)	−0.02 (−2.73 to 2.69)	0.99			
Cerebrovascular disease, (no = 0, yes = 1)	0.49 (−3.82 to 4.81)	0.82			
Peripheral artery disease, (no = 0, yes = 1)	0.05 (−5.02 to 5.11)	0.98			
Atrial Fibrillation, (no = 0, yes = 1)	−0.26 (−2.87 to 2.36)	0.84			
NYHA class (II–III), (no = 0, yes = 1)	3.63 (0.64 to 6.63)	0.018	1.99 (−0.61 to 4.59)	2.50	0.13
Obesity, (no = 0, yes = 1)	0.70 (−1.89 to 3.29)	0.59			
Dyslipidemia, (no = 0, yes = 1)	1.45 (−1.51 to 4.41)	0.33			
Diabetes, (no = 0, yes = 1)	−0.66 (−3.71 to 2.39)	0.67			
OSAHS, (no = 0, yes = 1)	−0.69 (−5.37 to 4.00)	0.77			
COPD, (no = 0, yes = 1)	−1.53 (−5.83 to 2.77)	0.48			
Anemia, (no = 0, yes = 1)	−0.25 (−3.49 to 2.98)	0.88			
Beta-blockers, (no = 0, yes = 1)	2.62 (−0.29 to 5.53)	0.08			
ACEI/ARB, (no = 0, yes = 1)	0.68 (−2.66 to 4.02)	0.68			
Diuretics, (no = 0, yes = 1)	1.30 (2.16 to 4.76)	0.46			
MR blockers, (no = 0, yes = 1)	1.80 (−0.77 to 4.37)	0.17			
Biomarker combination, (no = 0, yes = 1)	2.72 (0.20 to 5.24)	0.035	2.27 (0.19 to 4.34)	5.06	0.033

Abbreviations as in Table 1. * Variables with *p* ≤ 0.055 were selected for the multivariable analysis.

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
