# Peer review of "Does Chronic Kidney Disease Facilitate Malignant Myocardial Fibrosis in Heart Failure with Preserved Ejection Fraction of Hypertensive Origin?"

_jcm, 2020, doi:10.3390/jcm9020404_

Round 1
Reviewer 1 Report
The field of cardiorenal syndrome is vast, rapidly expanding in internal medicine. This field requires an interdisciplinary approach of a cardiologist and nephrologist. “Does chronic kidney disease facilitate malignant 2 myocardial fibrosis in heart failure with preserved 3 ejection fraction?” is an interesting article.
Minor revision:
If all patients was hypertension, maybe the title should be” Does chronic kidney disease facilitate malignant myocardial fibrosis in heart failure hypertension patients with preserved ejection fraction?” In the end of the sentence 39 citation should be in square bracket -dysfunction1. In the end of the sentence 52 citation should be in square bracket- HF3. In line 55 -4 should be in square bracket-causes4. In line 85 should be added that CKD is defined as (eGFR<60 ml/min/1.73m2) for 3 months or more. In material and methods should be information how blood pressure was measure. Also should be added that hypertension is was defined as… References should be numbered.
Author Response
Dear reviewer,
Thank you for your encouraging and insightful comments, which have helped us to improve the quality of our work. Following are the responses to your comments. For your convenience the changes introduced in the text in accordance with your suggestions are in red.
1) If all patients was hypertension, maybe the title should be” Does chronic kidney disease facilitate malignant myocardial fibrosis in heart failure hypertension patients with preserved ejection fraction?”
In agreement with the reviewer suggestion, we have changed the title to: “Does chronic kidney disease facilitate malignant myocardial fibrosis in heart failure with preserved ejection fraction of hypertensive origin?”
2) In the end of the sentence 39 citation should be in square bracket -dysfunction1. In the end of the sentence 52 citation should be in square bracket- HF3. In line 55 -4 should be in square bracket-causes4.
Following the reviewer’s indications, all these references are now enclosed in squared brackets.
3) In line 85 should be added that CKD is defined as (eGFR<60 ml/min/1.73m2) for 3 months or more.
Following the reviewer’s requirements, we have incorporated “..for ≥3 months” in line 93 (see page 3).
4) In material and methods should be information how blood pressure was measure. Also should be added that hypertension is was defined as…
As the reviewer requires, information on the blood pressure measurement protocol has been added in lines 80-85 (see page 2).
5) References should be numbered.
We have checked again that all references are numbered.
Reviewer 2 Report
The study by Eiros et al identifies the histomolecular phenotype of malignant myocardial fibrosis (high serum PICP and low serum CITP:MMP-1) is associated with CKD in hypertensive patients with HFPEF. This biomarker combination may potentially be able to select these types of patients for suitable anti-fibrotic therapy.
With relatively low numbers of patients in the group the study still identifies a novel indication for this biomarker combination which could lead to new treatments for this cohort. The methodology and science is very sound and well presented.
As the authors already identify there are several limitations which need to be addressed and validated in larger cohorts of patients; and which could be extended to other groups of HF patients in future.
The current study relies on the definition of mMF based on a previous study of hpertensive patients with HF based on pathological assessment of endomyocardial biopsies from the same patients. Future studies should examine this aspect in different disease states.
The wording in lines 306-309 states "Therefore, the excess of serum PICP in hypertensive patients with 306 HFpEF compared with hypertensive patients without HF here reported suggests that the myocardial synthesis and deposition of collagen type I fibers increases in parallel with the progression towards symptomatic HF in these patients". The word suggests is rather strong as collagen deposition was not measured. I suggest the authors soften this statement.
Some repetitiveness of the methods is in the results section
Author Response
Dear reviewer,
Thank you for your encouraging and insightful comments, which have helped us to improve the quality of our work. Following are the responses to your comments. For your convenience the changes introduced in the text in accordance with your suggestions are in blue.
1) The current study relies on the definition of mMF based on a previous study of hypertensive patients with HF based on pathological assessment of endomyocardial biopsies from the same patients. Future studies should examine this aspect in different disease states.
In agreement with the reviewer’s comment, the following sentence has been added to limitations: “….further studies should be performed to examine the correspondence between histological and biochemical aspects of MIF in different HF stages”. See lines 375-377 in page 14)
2) The wording in lines 306-309 states "Therefore, the excess of serum PICP in hypertensive patients with 306 HFpEF compared with hypertensive patients without HF here reported suggests that the myocardial synthesis and deposition of collagen type I fibers increases in parallel with the progression towards symptomatic HF in these patients". The word suggests is rather strong as collagen deposition was not measured. I suggest the authors soften this statement.
As required by the reviewer, we have toned down the sentence in lines 313-316 (page 13) to the following: “…it is tempting to speculate that the excess of serum PICP in hypertensive patients with HFpEF compared with hypertensive patients without HF here reported could reflect an increase in myocardial synthesis and deposition of collagen type I fibers in parallel with the progression towards symptomatic HF in these patients”
3) Some repetitiveness of the methods is in the results section
The reviewer is right. We have deleted a sentence in results already explained in the methods section (see lines 222-224 in page 8).